# Geochemical Referencing of Natural Forest Contamination in Poland

**Paweł Rutkowski [1,\*], Jean Diatta [2], Monika Konatowska [1], Agnieszka Andrzejewska [2], Łukasz Tyburski [3] and Paweł Przybylski [4]**

[1] Department of Forest Sites and Ecology; Faculty of Forestry, Poznań University of Life Sciences, Wojska Polskiego 71F, 60-625 Poznań, Poland; monika.k-poznan@wp.pl
[2] Department of Agricultural Chemistry and Environmental Biogeochemistry, Faculty of Agriculture and Bioengineering, Poznan University of Life Sciences, Wojska Polskiego 71F, 60-625 Poznań, Poland; jeandiatta63@yahoo.com (J.D.); agnieszka.andrzejewska11@gmail.com (A.A.)
[3] Kampinoski National Park, Tetmajera 38, 05-080 Izabelin, Poland; ltyburski@kampinoski-pn.gov.pl
[4] Department of Silviculture and Forest Tree Genetics, Forest Research Institute, Braci Leśnej 3, Sękocin Stary, 05-090 Raszyn; p.przybylski@ibles.waw.pl
\* Correspondence: redebede@wp.pl

**Abstract:** Various studies have established possible threats posed due to pollution using ecological risk indices, but most have focused strictly on anthropogenic areas, so the data from these studies are less comparable with those obtained from natural forest sites, which was the focus of this current study. The main reason for this focus could be attributed to the commonly agreed reference provided by natural forest parks, which are assumed to be uncontaminated. The aim of this research was to determine if the Kampinoski National Park (Poland) could be considered a geochemical referencing ecosystem for Pb, Cd, and Ni levels. The specific purpose was to conduct a soil-background-based evaluation of metal contamination with a focus on geochemical indices as normative tools for assessing similar forest ecosystems at local and international levels. The toxicity response factors indicated some specific metal features that seemed highly magnified for Cd compared with Pb and Ni. The use of geochemical indices when assessing the contamination status of various ecosystems, either natural or strongly anthropogenic, is recommended to enable worldwide comparison, rather than only assessing metal contents. This approach considers the background metal concentrations for local on-site targets as well as pre-industrial reference levels for international referencing.

**Keywords:** ecological risk index; geo-accumulation; natural forest; referencing

## 1. Introduction

The introduction of large amounts of chemical substances into the environment by human activity has caused enrichment of many elements in surface waters, sediments, and soils, particularly in industrial areas [1,2]. Studying these disturbed ecosystems—called anthropogenic anomalies—has become an essential part of geochemical and environmental research [3–14].

The term "geochemical background" (GB) [1,8,15–19] is one of the key notions for investigating environmental conditions. Distinguishing between natural levels of a given substance and those resulting from human activity can be challenging, especially when determining the level of environment pollution [20]. The principles and main methods used for the determination of background values are outlined in ISO 19258 (2018 and earlier versions). As the Swedish version of this document says [21], in practice, distinguishing clearly between the pedo-geochemical and the anthropogenic fraction of the background content of soils can be difficult. A detailed knowledge of the background content as

well as of its natural fraction of the substances of concern is essential for both any evaluation of the current status of the soil for environmental- or land-use-related aspects and scientific purposes within the scope of pedology or geochemistry.

Contamination in soils by harmful elements and substances is an unavoidable consequence of industrialization and the development of civilization [22,23]. For example, in the 1980s, spruce stands (*Picea abies*) in Southwestern Poland suffered mass death as a consequence of the impacts of heavy industry. Ecological disasters have occurred in other parts of the country. Consequently, since 1989, the degree of forest damage in Poland has been assessed annually as part of the Forest Monitoring program, which is one aspect of the State Environmental Monitoring system; the assessment is based on the level of defoliation. Zajączkowski et al. [24] noted that forests participate in heavy metal air purification, but did not indicate whether this process has any impact on the concentration of pollutants in the soil. So, the aim of this research was to determine whether the Kampinoski National Park (KNP) could be considered a geochemical referencing ecosystem for lead, cadmium, and nickel levels. The specific purpose of the research was a soil-background-based evaluation of metal contamination with a focus on geochemical indices as normative broad tools for assessing similar forest ecosystems at local and international levels.

## 2. Research Area

Kampinoski National Park is the second largest national park in Poland. It is located near to Warsaw, the biggest Polish city. The highly urbanized area is located east and southeast of the park (Figure 1). In the west, which is the dominant wind direction, and in the southwest, the city of Łódź is located within a radius of 150 km, with a population of nearly 700,000 and a dispersed arrangement of smaller towns (Figure 1). Two parks, the Gostynin-Włocławek Landscape Park and the Bolimów Landscape Park, serve as buffer zones for the KNP (Figure 1). Two highways, the A2, established in 2012, and the A1, established in 2016, may also impact the KNP.

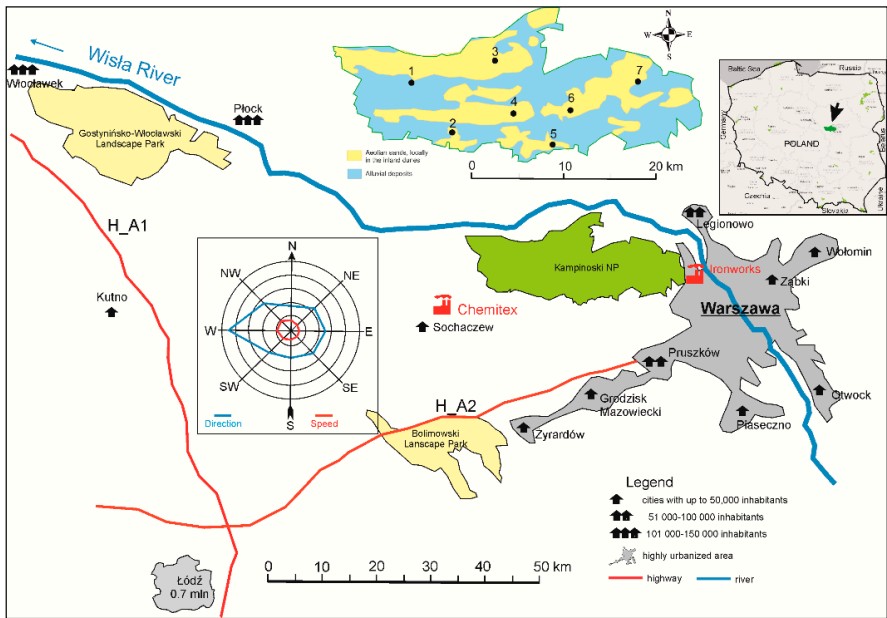

**Figure 1.** Location of Kampinoski National Park (KNP, area marked in green) in relation to the important spatial elements that may affect the park. In the upper right corner, the location of the KNP is marked against the background of the map of Poland. In the upper middle, a map of KNP is shown with the location of research plots (black spots) against the background of geological formations (yellow, aeolian sands; blue, alluvial deposits). Red icons on the main image indicate the most important possible sources of heavy metal emissions (chemical fiber plant in Chodaków on the left and smelter plant in Warsaw on the right).

In 2000, the KNP was included in the UNESCO list of world biosphere reserves. Among its many natural values, the most preserved, at the European scale, complex of inland dunes deserves special attention.

The combined values of the park were the determining factors for selecting of the seven study plots, which were located in dune areas (Figure 1) with pine stands older than 100 years. At each of the study sites, a soil pit was established. Each soil pit was 2 m long and 0.5 m deep. The axis of each soil pit was located in the north–south direction. From each soil pit, we sampled the litter layer and samples from depth: 5–10, 25–30, and 45–50 cm. The soil samples were placed on a canvas sheet on the western side of the pit and, after completion of the study, placed back into the pit. The eastern side remained intact [25].

## 3. Materials and Methods

### 3.1. Chemical Analysis of Pseudo-Total Content of Pb, Cd, Ni in Soil

The pseudo-total contents of Pb, Cd, and Ni in the soil were assayed according to analytical procedures reported by Ciesielski et al. [26] and Gupta et al. [27]. Some modifications were incorporated, which consisted of using a 6 mol HCl dm$^{-3}$ (ratio soil to extractant of 1:10), and then heating the slurry on a sand bath at 140 °C for 2 hours under reflux. All extractions were replicated twice. After cooling, the extraction was filtered through filter paper into 20 cm$^3$ test tubes and filled up to the mark with bi-distilled water.

The concentrations of Pb, Cd, and Ni were determined by atomic absorption spectrometry (AAS; Varian SpectrAA 250 plus, Varian Inc., Palo Alto, CA, USA). The relative standard deviation (RSD) was calculated from pooled data for applied methods. In the precision test, the average RSD (%) for all metals ranged from 0.75% to 0.95%. The accuracy of the pseudo-total metal contents was determined using a reference material (Estuarine sediment 277 CRM certified by the Bureau Community of Reference (BCR), Brussels, Belgium). Particle size was analyzed according to the Bouyoucos method modified by Casagrande and Proszynski (aerometric method) for fractions <0.1 mm in soil mass. Fractions >0.1 mm were analyzed on sieves with the following mesh sizes: 0.1, 0.25, 0.5, 1.0, and 2.0. The dominant particle size, in all tested soils, was 0.25–0.5 mm.

### 3.2. Geochemical Indices for Evaluating Heavy Metal Contamination

The use of geochemical indices when assessing the contamination status of various ecosystems (natural and strongly anthropogenic) is recommended to enable worldwide comparison compared to comparison based on metal contents alone [8,22,28–31]. This approach considers background metal concentrations (for local on-site targets) as well as pre-industrial reference levels (for international referencing).

### 3.3. Geo-Accumulation Index

The geo-accumulation index ($I_{geo}$) is used for outlying and quantifying the degree of anthropogenically or geogenically accumulated pollutants in environmental sites [28]. We used $I_{geo}$ to evaluate Pb, Cd, and Ni concentrations and hence potential pollution in the investigated natural forest ecosystem. The formula used for this purpose was:

$$I_{geo} = \log_{10} \frac{C_n}{1.5\, B_n} \tag{1}$$

where $C_n$ is the metal concentration in the soil, $B_n$ is the pre-industrial reference [29], the values of which are listed in Table 1 along with geochemical background values [32]. The value 1.5 expresses natural fluctuations in the concentrations of a given metal within an environment under slight anthropogenic influence.

**Table 1.** Geochemical background, pre-industrial reference levels, and toxic response factors.

| Metal | Czarnowska | Håkanson | |
|---|---|---|---|
| | Geochemical Background | Pre-Industrial Reference Level | Toxic Response Factor |
| | mg kg$^{-1}$ | | - |
| Lead (Pb) | 9.8 | 7.0 | 5.0 |
| Cadmium (Cd) | 0.18 | 1.0 | 30 |
| Nickel (Ni) | 10.2 | 20 | 5.0 |

The $I_{geo}$ index categorizes seven classes of contamination of environmental samples as listed in Table 2 [29,33].

**Table 2.** Classification of geo-accumulation index and respective description.

| Contamination Class | Index Value | Contamination Status |
|---|---|---|
| 0 | $I_{geo} \leq 0$ | No contamination |
| 1 | $0 < I_{geo} < 1$ | Slight to moderate contamination |
| 2 | $1 < I_{geo} < 2$ | Moderate contamination |
| 3 | $2 < I_{geo} < 3$ | Moderate to strong contamination |
| 4 | $3 < I_{geo} < 4$ | Strong contamination |
| 5 | $4 < I_{geo} < 5$ | Strong to extremely serious contamination |
| 6 | $5 < I_{geo}$ | Extremely serious contamination |

*3.4. Contamination Factor and Contamination Degree*

The contamination factor ($C_f^i$) is also called the single pollution index [29], which expresses the quotient obtained by dividing the concentration of metals related to the investigated site by the reference or background values. It is calculated as:

$$C_f^i = \frac{C_{0-1}^i}{C_n^i} \qquad (2)$$

where $C_{0-1}^i$ is the metal content of the investigated site and $C_n^i$ is the pre-industrial reference [29] and geochemical background [32] values (Table 1).

The contamination degree ($C_{deg}$) is the sum of all contamination factors ($C_f^i$) and is reported in Table 3:

$$C_{Deg} = \sum C_f^i \qquad (3)$$

where $C_{deg}$ is the contamination degree.

**Table 3.** Contamination factor and degree and their respective description.

| Contamination Factor | | Contamination Degree | |
|---|---|---|---|
| Value | Description | Value | Description |
| $C_f^i < 1$ | Low contamination factor | $C_{Deg} < 7$ | Low degree of contamination |
| $1 < C_f^i < 3$ | Moderate contamination factor | $7 < C_{Deg} < 14$ | Moderate degree of contamination |
| $3 < C_f^i < 6$ | Considerable contamination factor | $14 < C_{Deg} < 21$ | High degree of contamination |
| $6 \geq C_f^i$ | Very high contamination factor | $C_{Deg} \geq 21$ | Very high degree of contamination |

## 3.5. Ecological Risk Index (ERI)

We used the ecological risk index ($ERI_r{}^i$) for quantitatively expressing the potential ecological risk of investigated metals, as suggested by Håkanson [29]:

$$ERI_r{}^i = Tr^i \cdot C^i{}_f \tag{4}$$

where $Tr^i$ is the toxic response factor for the metals and $C^i{}_f$ is the contamination factor. The $Tr^i$ values of heavy metals reported by Håkanson [29], Qing et al. [34], and Wu et al. [35] are listed in Table 1. Recommendations for the use of the ERI are outlined in Table 4. Although the risk index was originally used as a diagnostic tool for the purpose of controlling water pollution, it was successfully used for assessing the quality of sediments and soils at various ecological sites [36].

**Table 4.** Grades of the ecological (ERI) and potential ecological risk index (PERI) [37].

| Grade | ERI Value | ERI Grade of Single Metal | PERI Value | Environmental PERI Grade |
|-------|-----------|---------------------------|------------|--------------------------|
| A | $ERI_r{}^i < 5$ | Low risk | $PERI < 30$ | Low risk |
| B | $5 \leq ERI_r{}^i < 10$ | Moderate risk | $30 \leq PERI < 60$ | Moderate risk |
| C | $10 \leq ERI_r{}^i < 20$ | Considerable risk | $60 \leq PERI < 120$ | Considerable risk |
| D | $20 \leq ERI_r{}^i < 40$ | High risk | $PERI \geq 120$ | Very high risk |
| E | $ERI_r{}^i \geq 40$ | Very high risk | - | - |

## 3.6. Potential Ecological Risk Index (PERI)

The potential ecological risk index (PERI) is defined as the sum of the risk factors, as is the degree of contamination:

$$PERI = \sum_{i=1}^{m} ERI_r^i \tag{5}$$

where $ERI^i$ is a single ecological risk index and $m$ is the number of heavy metal species. Grades for the PERI are reported in Table 4 [29].

## 4. Results

### 4.1. Quantitative-Based Contamination Evaluation

Natural forest ecosystems are facing increasing anthropogenic pressure resulting from various sources such as forest stand management, tourism and site seeing, and wet and dry depositions from industrial fallouts. In terms of contamination impacts, the latter appears much more quantifiable, particularly when considering anthropogenic heavy metals. The common challenge encountered by environmentalists emerges at the quantitative (amount or concentration basis, Table 1) or the qualitative evaluation (index basis, Tables 2–4) of the state of cleanness of the natural ecosystem.

We decided to evaluate the contamination state of the Kampinoski National Park by applying two sources of data: local geochemical background value [32] and worldwide reference value [29], (Table 1). This approach provided the opportunity to integrate both criteria and particularly outline the tasks and targets required. We needed to determine if the discrepancies in the background values were significant enough to impede the evaluation, given differences between background and measured values reaching 29% for Pb, 82% for Cd, and 49% for Ni. We wanted to know which of these values could considered for referencing on-site.

Data listed in Table 5 outline the wide heavy metal content variability within the park sites, specifically for Pb (37.5%) and Ni (36.5%), compared to Cd (13.1%). The number of sampling sites (Figure 1) was large and disparate enough to capture any variability, which could be attributed to two reasons: deposition of anthropogenic emissions initiated by the wind and natural geochemical

concentrations of these metals. At all sites, Pb levels exceeded the background contents (Table 1), but Ni showed a reverse pattern: its site contents were lower compared to the background values. The pre-industrial reference level (20.0 mg kg$^{-1}$) [29] was two times higher than that suggested by Czarnowska [32] and four times higher with respect to the mean value (5.11 mg kg$^{-1}$) for the whole park. Cd contents varied within the range of 0.21 to 0.32 mg kg$^{-1}$, which are higher than 0.18 mg kg$^{-1}$ and lower compared to 1.0 mg kg$^{-1}$, respectively (Table 1). At this stage, any quantitative evaluation of park contamination is strictly related to these values, but Cd-based decision making seems fairly convincing.

**Table 5.** Pseudo-total heavy metal contents at the Kampinoski National Park for the whole 5–50 cm depth sampling layer.

| Sampling location | Pb | Cd | Ni |
|---|---|---|---|
| | mg kg$^{-1}$ | | |
| Site 1 | 29.2 | 0.28 | 5.60 |
| Site 2 | 22.7 | 0.29 | 4.56 |
| Site 3 | 20.6 | 0.21 | 3.80 |
| Site 4 | 13.8 | 0.32 | 4.81 |
| Site 5 | 12.4 | 0.32 | 9.07 |
| Site 6 | 32.7 | 0.27 | 4.16 |
| Site 7 | 15.1 | 0.30 | 3.74 |
| Mean ± SD | 20.9 ± 7.8 | 0.28 ± 0.04 | 5.11 ± 1.86 |
| CV(%) [a] | 37.5 | 13.1 | 36.5 |

[a] Coefficient of variation.

### 4.2. Indices-Based Contamination Evaluation

#### 4.2.1. Geo-Accumulation Index

The indices listed in Table 6 and illustrated in detail in Figure 2 were calculated using pre-industrial [29] and geochemical background [32] values using Equation (1). Next, the recorded indices were subjected to evaluation criteria according to Table 2. We found that the entire park was categorized as class 0 (zero) meaning no contamination by Pb ($I_{geo}$-Pb < 0). This contamination state was quite similar at all locations irrespective of background values. Data from Figure 2 show that the relatively lower Pb content [29] resulted in slightly higher $I_{geo}$-Pb values compared to those reported by Czarnowska [32].

**Table 6.** Geo-accumulation indices for the particular heavy metals throughout Kampinoski National Park (7 sites, Figure 1).

| Metal | Background values by | Geo-accumulation index | Min | Max | Mean ± SD (*n* = 7) |
|---|---|---|---|---|---|
| Pb | | | 0.42 | 0.85 | 0.63 ± 0.16 |
| Cd | Håkanson | $I_{geo}$-H | –0.49 | –0.32 | –0.37 ± 0.06 |
| Ni | | | –0.55 | –0.17 | –0.44 ± 0.13 |
| Pb | | | 0.28 | 0.70 | 0.48 ± 0.16 |
| Cd | Czarnowska | $I_{geo}$-C | 0.25 | 0.43 | 0.37 ± 0.06 |
| Ni | | | –0.26 | -0.08 | –0.19 ± 0.06 |

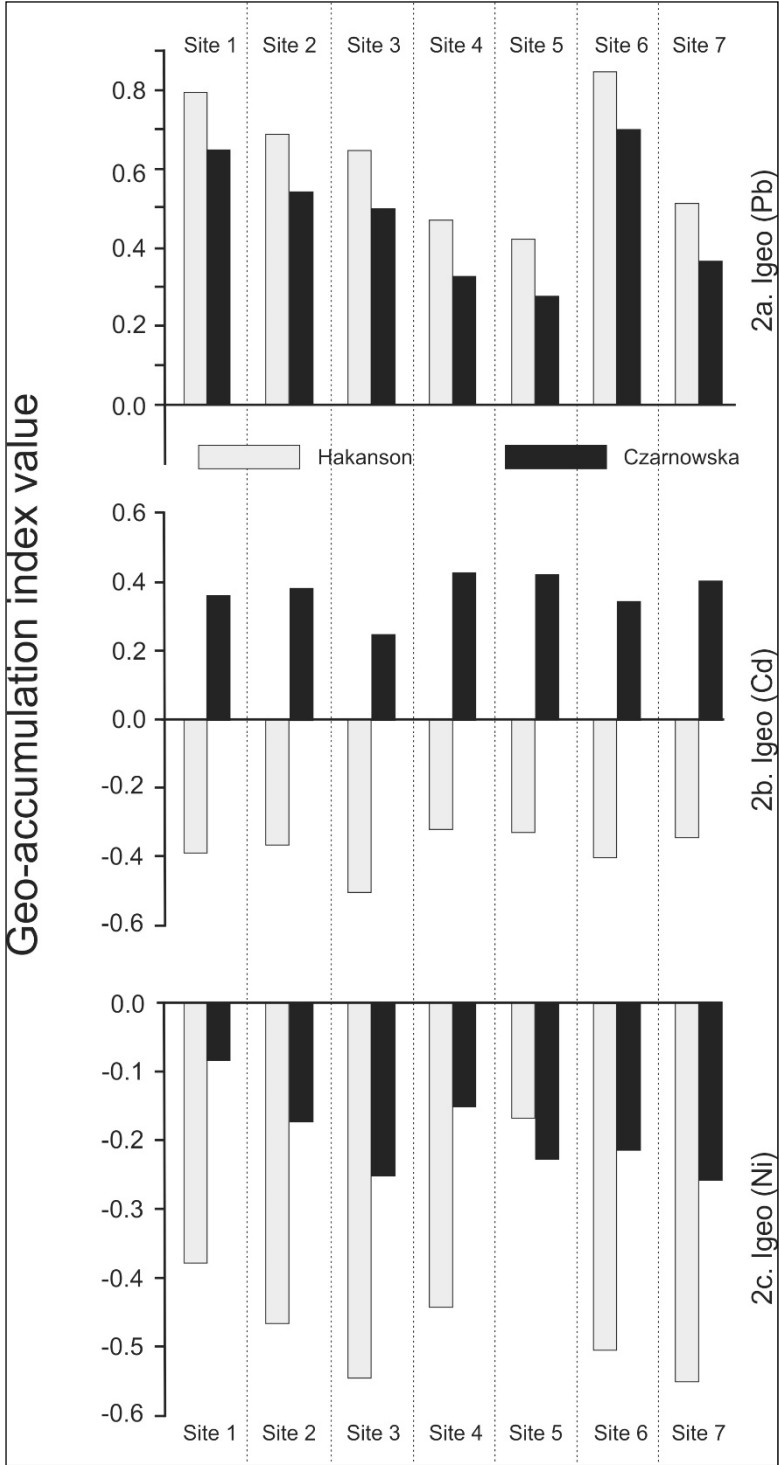

**Figure 2.** Site distribution of (a) Pb, (b) Cd, (c) and Ni geo-accumulation indices within the Kampinoski National Park ($I_{geo}$ assayed on the basis of Håkanson and Czarnowska background values, Table 1).

The spread of these indices is site-specific since the pattern decreased progressively from sites 1 to 4, where no significant alteration could be observed, except for at site 6. This spatial distribution of Pb may be linked to the wind (Figure 1) and then possibly to anthropogenic sources.

The case of Cd geo-accumulation indices ($I_{geo}$-Cd) deserves some attention, since sampling sites could be clearly differentiated by background values compared with the Cd content on-site. Mean $I_{geo}$-Cd values (Table 6) varied widely: between –0.37 and 0.37 compared with the background values

reported by Håkanson and Czarnowska, respectively. The overall Pb evaluation implies that class 0 (no contamination) prevailed at all sites. The approximately six-fold difference in the background values (0.18 versus 1.0 mg kg$^{-1}$) could cause some confusion when establishing forest soil quality regulations. Then, $I_{geo}$-Cd values could support some very strict restrictions related to contamination regulation when calculated on the basis of 0.18 compared to 1.0 mg kg$^{-1}$.

Irrespective of the possibilities indicated by $I_{geo}$-Cd site classifications, we found that the Kampinoski National Park is not contaminated by cadmium.

Nickel was the third metal evaluated in terms of site contamination. Its pseudo-total content (Table 5) did not exceed the background values, so the area should be classified as uncontaminated. The $I_{geo}$-Ni indices (Table 6) support this finding since the mean values fluctuated between –0.14 and –0.44 compared to the background values reported by Håkanson and Czarnowska, respectively. This is illustrated in Figure 2, where the trend is slightly consistent with that observed for $I_{geo}$-Pb, i.e., in line with the wind main directions. Indices slightly and progressively decreased from sites 1 to 7 (except site 5). The reported spatial distribution pattern does not mean that the whole forest park ecosystem is uncontaminated (mean $I_{geo}$-Ni indices –0.37 and –0.44 for Håkanson and Czarnowska background values, respectively). Similar for $I_{geo}$-Cd indices, the two-fold difference compared to the background values (10.2 versus 20 mg kg$^{-1}$) may be support stringent restrictions when the value 10.2 mg kg$^{-1}$ is applied; if 20.0 mg kg$^{-1}$ is applied, moderate restrictions on emissions are supported.

### 4.2.2. Single and Joint Effects of Heavy Metal Contamination

KNP contamination could also be evaluated by assessing the impacts of Pb, Cd, and Ni individually or in combination. As such, we used additional indices such as contamination factors ($C_f^i$) and degree of contamination ($C_{deg}$) for processing experimental data, and the results are summarized in Table 7 and illustrated in Figure 3 ($C_{deg}$) for the whole investigated area. The contamination factors varied widely according to the background values used for their calculation. In the case of the Håkanson values, the mean metal-based $C_f^i$ order was: Pb (2.99) > Ni (0.50) > Cd (0.29); whereas for Czarnowska's values, the pattern was Pb (2.14) > Cd (1.58) > Ni (0.26). The criteria listed in the Table 3 for performing the ground evaluation of the investigated area revealed that two ranges prevailed: $C_f^i <$ 1 (low contamination) and 1 $< C_f^i <$ 3 (moderate contamination). The contamination state attributed to the effect of Ni was low for the whole park; the same applied to Cd when evaluated on the basis of the Håkanson background values. The same results were found for Pb.

**Table 7.** Contamination factors and degrees for the particular heavy metals for the entire Kampinoski National Park (7 sites, Figure 1).

| Metal | Contamination factor | | | Mean ± SD | Share (%) $C_f^i$ in $C_{deg}$ |
|---|---|---|---|---|---|
| | $C_f^i$ by | Min | Max | | |
| Pb | | 1.77 | 4.68 | 2.99 ± 1.12 | 79.1 |
| Cd | Håkanson | 0.21 | 0.32 | 0.29 ± 0.037 | 7.7 |
| Ni | | 0.37 | 0.89 | 0.50 ± 0.18 | 13.2 |
| Contamination degree [$C_{deg}= \Sigma(C_f^{Pb, Cd, Ni})$] | | 2.77 | 5.35 | 3.78 ± 1.05 | - |
| Pb | | 1.26 | 3.34 | 2.14 ± 0.80 | 53.8 |
| Cd | Czarnowska | 1.19 | 1.79 | 1.58 ± 0.21 | 39.7 |
| Ni | | 0.19 | 0.45 | 0.26 ± 0.093 | 6.5 |
| Contamination degree [$C_{deg}= \Sigma(C_f^{Pb, Cd, Ni})$] | | 3.43 | 5.04 | 3.98 ± 0.70 | - |

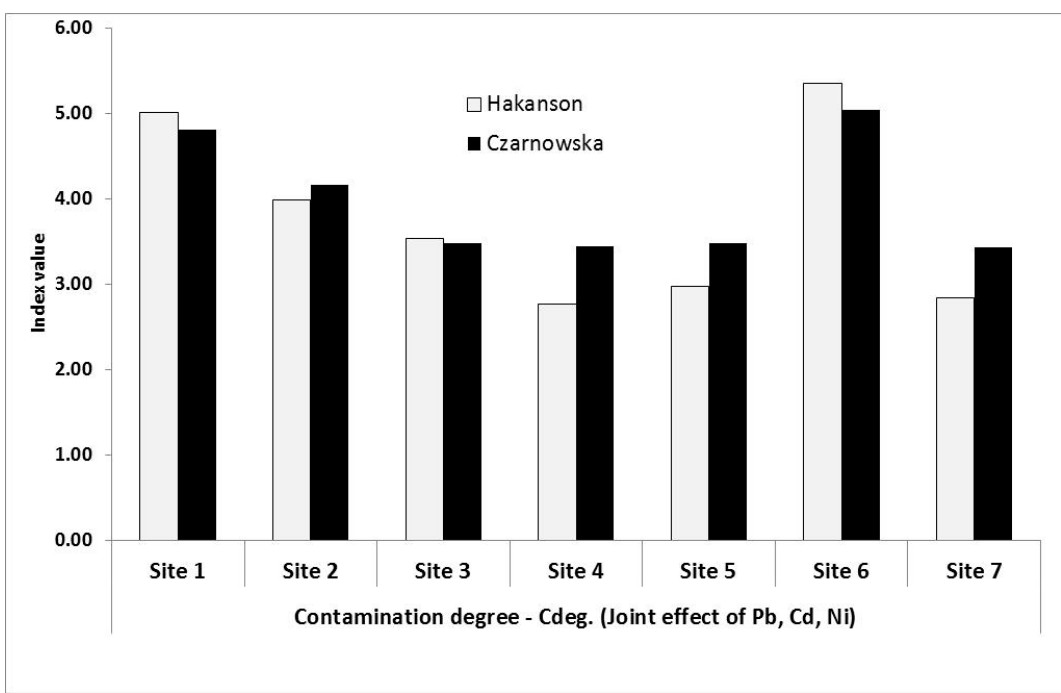

**Figure 3.** Site distribution of contamination degree indices ($C_{deg}$) within the Kampinoski National Park ($C_{deg}$ assayed on the basis of Håkanson and Czarnowska background values, Table 1).

On the basis of the criteria reported in Table 3, it we found that the investigated area should be described as moderately contaminated.

The environmental response to the impacts of trace metals and heavy metals, among others, is generally driven by the complex nature of the metals. Therefore, the use of indices such as degree of contamination ($C_{deg}$) for a global evaluation of the site should more accurately depict the situation than considering metals individually. We considered the range $C_{deg} < 7$, indicating a low degree of contamination. None of the calculated indices exceeded this threshold, and mean values were 3.78 and 3.98 established from the Håkanson and Czarnowska background values, respectively. We found that Pb could be identified as a discrete contamination-driven metal, since it contributed over 50% of the overall park contamination (Table 7).

The trend displayed by $C_{deg}$ indices (Figure 3) followed a pattern consistent with that observed for $I_{geo}$-Pb, and somewhat with that of $I_{geo}$-Ni. This implies the lack of metal-related contamination threats and the natural ecosystem of the whole park is still intact. However, as indicated by Table 7, all investigated sites were more or less under slightly high or slightly low effects of Pb. In other words, wind characteristics may have produced this situation; hence, industrial expansion and further activities on the western side of the KNP should be maintained with high environmental standards, particularly concerning emissions fallouts and their subsequent wet or dry deposition.

### 4.2.3. Ecological Risk Indices

We more accurately assessed heavy metal effects and their relevant ecological risks (ERI) as well as potential ecological risk (PERI) indices to determine the real threat to the KNP. These indices were calculated by considering contamination factors ($C_f^i$) and the values of the toxicity response (Tr$^i$) of the environment (Table 1). The PERI was calculated as the sum of individual risk indices (ERI). Data for both indices are reported in Table 8 and illustrated in Figure 4. The individual and complex evaluation was performed on the basis of grades listed in Table 4.

**Table 8.** Ecological risk index ($ERI_r^i$) and PERI for the particular heavy metals at the whole NP (7 sites, Figure 1).

| Metal | Ecological risk index | | | Mean ± SD (n = 7) | Share (%) $ERI_r^i$ in PERI |
| --- | --- | --- | --- | --- | --- |
| | $ERI_r^i$ | Min | Max | | |
| Pb | Håkanson (1980) | 8.8 | 23.3 | 15.0 ± 5.6 | 57.5 |
| Cd | | 6.4 | 9.7 | 8.6 ± 1.1 | 32.9 |
| Ni | | 1.8 | 4.4 | 2.5 ± 0.9 | 9.6 |
| Potential ecological risk index $PERI = \Sigma(ERI^{Pb,Cd,Ni})$ | 21.8 | 33.5 | 26.0 ± 4.9 | - |
| Pb | Czarnowska (1996) | 6.3 | 16.7 | 10.7 ± 4.0 | 17.9 |
| Cd | | 35.7 | 53.8 | 47.5 ± 6.2 | 79.9 |
| Ni | | 0.9 | 2.3 | 1.3 ± 0.5 | 2.2 |
| Potential ecological risk index $PERI = \Sigma(ERI^{Pb,Cd,Ni})$ | 47.1 | 62.5 | 59.5 ± 5.6 | - |

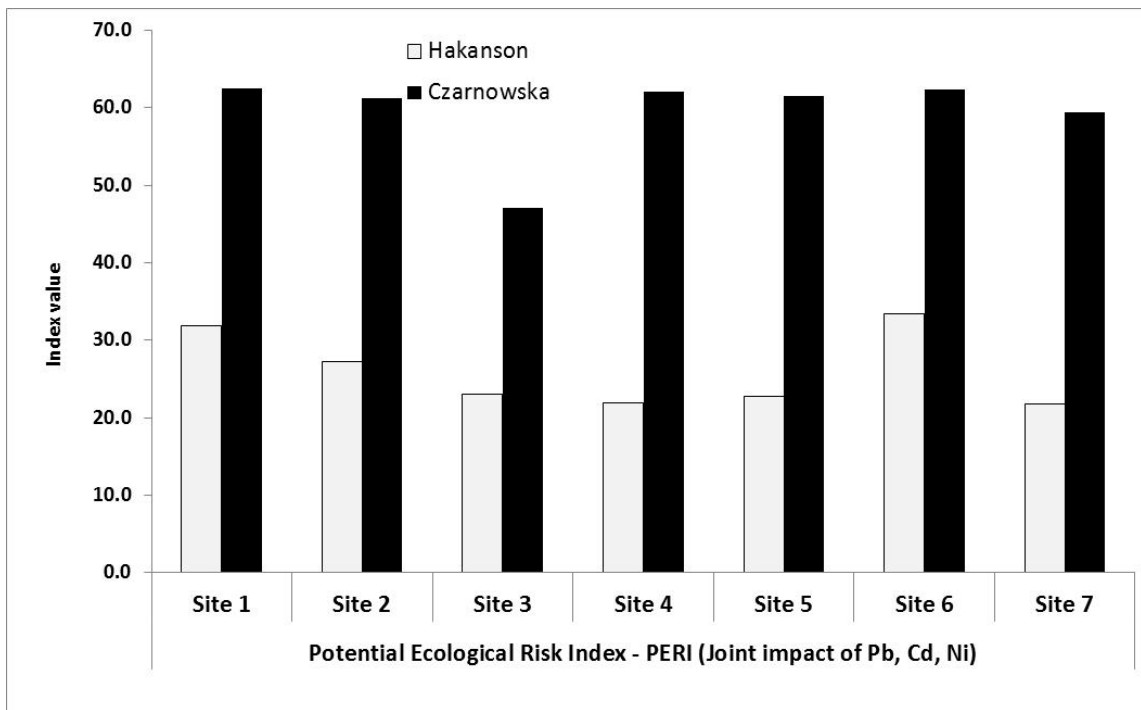

**Figure 4.** Site distribution of PERI within the Kampinoski National Park (PERI assayed on the basis of Håkanson and Czarnowska background values, Table 1).

The ERI mean values varied with heavy metal contents on-site as well as the background values used for calculating the indices. A larger range was observed for the Czarnowska values, 1.28 to 47.52, as compared to the narrower range using Håkanson's values of 2.50 to 4.96. Two metals were identified as significantly involved in shaping the ERIs: Pb and Cd. For lead and cadmium, their values in both cases were within the range of 10.68 to 14.96 (grade: considerable risk) and 8.55 to 47.52 (grade: moderate to very high risk), respectively. The consideration of the toxicity response factor (Tr) identified some specific metal features, which were more highly magnified for Cd (Tr = 30).

The share of ERI-Pb and ERI-Cd in the PERI fluctuated in widely, where the lowest value was related to Ni (both for Håkanson and Czarnowska background values). Lead and cadmium individually shared according to their ERI values, raising then the ecological risk awareness within the KNP.

A much more comprehensive and integrated evaluation of the ecological state was performed on the basis of the potential ecological risk index (PERI), whose values (grades) are synthetically listed

in Table 8 and illustrated in Figure 4. The criteria applicable to the ground assessment of the entire KNP are outlined in Table 4. On this basis, we observed that the reported mean PERI values—i.e., 26.02 and 59.48—fitted closely the grade range 30 < PERI < 60, indicating a moderate ecological risk. Our key finding is that PERI values from Figure 4 are almost spread equally (inconsistent with the wind, Figure 1) within the park, implying that the natural forest ecosystem has some mechanisms to mitigate as well as counteract anthropogenic impacts (threats) of Pb, Cd, and Ni.

## 5. Discussion

The natural value of the Kampinoski National Park may be basically categorized into flora and fauna. However, this approach lacks several considerations, like being the second greatest national park in Poland, a UNESCO biosphere reserve, and an important area on the European scale of X35 habitat (Inland Sand Dunes), listed in the Convention on the Conservation of European Wildlife and Natural Habitats. The forest protects one of the oldest pine stands in Poland (some >200 years old), supplies fresh air to the Warsaw agglomeration (according to wind direction), and accommodates around 1 million visitors annually with over 500 km of various trails. A rough analysis of the last factor indicates that such anthropogenic pressure could result in contamination of the park. This statement is strongly time- and space-limited, since only a limited area of the park is reserved for visitors and mostly during the summer.

The question arises of any possible stratification of metal content in the soil within the KNP. This is an important factor influencing contamination magnification as well as dissemination at respective sampling sites. The data summarized in Table 9 outline a specific metal distribution at two key forest layers: litter (5–0 cm) and organic-mineral soil layer (−5 to −10 cm). The highest percentage of heavy metal content in the litter was recorded for Cd (73.0%) followed by Pb and Ni equally (ca. 50%). SD values were similar for Cd and Ni, but much higher, i.e., 16.2%, for Pb. This disparity results strictly from the sampling sites as illustrated in Figure 5, where the elevated share of Pb can be attributed to its new inputs originating from the southeast (Warsaw agglomeration) and southwest (highway A2) to the park (Figure 1). When the layer extends to −5 to −10 cm, no significant discrepancies (SD = 4.6%) emerged between any of the sites. This implies that the 5–0 cm layer litter is undergoing dynamic alterations in Pb accumulation within the KNP, but the sources could have a relatively short time action and are more likely attributable to traffic intensity. In this case, the highway A2 should be better controlled and other options provided to minimize its use.

**Table 9.** Percentage share of Pb, Cd, and Ni in the 5–0 and 5–10 cm layers of the whole (5 to –50 cm) sampling depth.

| Metal share in respective layers | | Pb | Cd | Ni |
|---|---|---|---|---|
| | | % ± SD | | |
| 5–0 cm | 5 to –50 cm | 49.6 ± 16.2 [a] | 73.0 ± 6.9 | 50.0 ± 6.2 |
| −5 to −10 cm | | 87.2 ± 4.6 | 84.6 ± 6.0 | 61.7 ± 7.1 |
| Δ = (−5 to −10 cm) − (5–0 cm) | | 37.6 | 11.6 | 11.7 |

[a] relatively high value of SD (details in Figure 5).

Another specific finding (Table 9) was the high share of all metals in the −5 to −10 cm layer in the whole (5 to –50 cm) sampling depth, characterized by a slight variation (SD = 4.6%–7.1%). This observation shows that: (1) for ephemeral evaluation of metal accumulation/contamination, the 5–0 cm layer should be examined, and (2) for perennial traceability of metal accumulation/contamination, the −5 to −10 cm layer should be prioritized.

The metal share was found to be, in decreasing order, as follows: Pb > Cd > Ni, with increases discriminating other metals from Pb (Δ = 37.6%), compared to Cd and Ni for ca. Δ = 11.6%. This implies that the investigated metals are not of geogenic origin but were produced from anthropogenic

sources. Notably, the 5 to –10 cm layer operates as a buffering shield between the mineral subsoil and external litter atmosphere.

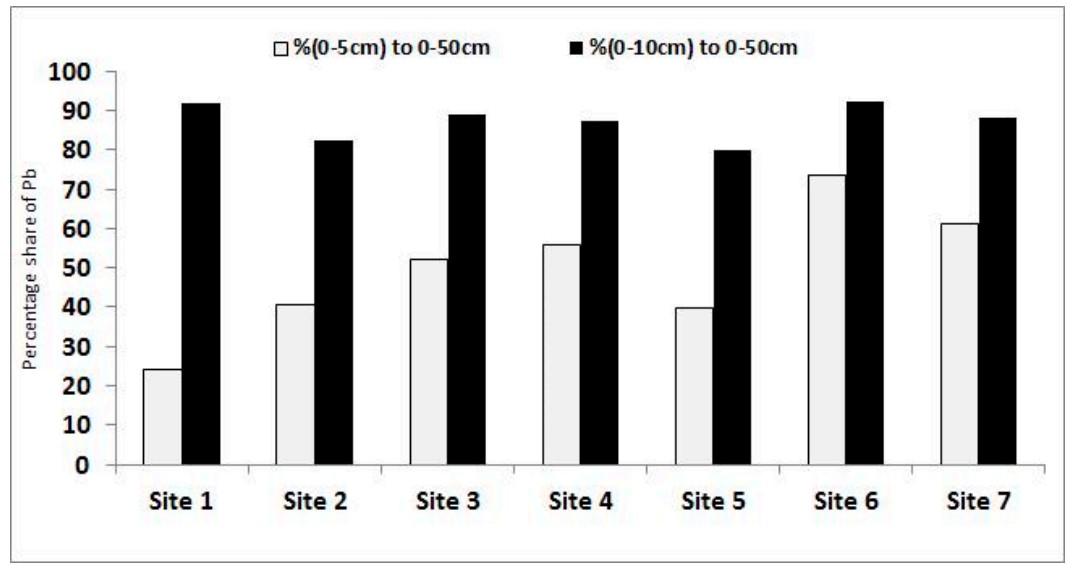

**Figure 5.** Percentage share of Pb from the 5–0 (litter) and −5 to −10 cm layers in the whole (5 to –50 cm) sampling depth.

Detailed studies of heavy metal content in the KNP carried out in 1976–1980 showed that in sandy soils, the variation was small in the amount of heavy metals in the soil profile, except for the organic layer. The researchers [38] found a relationship between the content of heavy metals in the topsoil and the distance of the sampling site from the source of pollution. Soils located within the pollution transport range contained more Zn, Pb, and Cu than soils distant from the source of pollution. Comparative studies with forest soils from the outskirts of Warsaw showed a higher level of heavy metals in the litter layer. With the exception of Cd, whose content was similar in soils in the immediate vicinity of Warsaw and KNP in the top layer of forest soil, we found= that Cd pollution is the result of air pollution.

The search for any contamination threat is necessary to protect the flora and fauna dwelling this precious environment, as well as humans. Therefore, we constructed a complex but easily implementable methodology to evaluate the reliability of referencing metal effects in natural forest ecosystems. The parameters and indices used for outlying the specific contamination process revealed some challenging factors that must be considered: background values, type of heavy metals, and characteristics of the site (i.e., location compared with urban and anthropogenic/industrial sources of emissions).

Various studies have been undertaken to establish possible threats due to pollution using ecological risk indices [39–44]. However, the majority of studies focused strictly on anthropogenic areas, decreasing the comparability of the data with those obtained from natural forest sites, as was the case here. The main reason for the anthropogenic focus of previous studies could be the commonly agreed assumption that Natural Forest Parks are not contaminated. A study by Mazurek et al. [44] on the Roztocze National Park forest soils (SE Poland) showed that pollution indices revealed Pb, Zn, Cu, and Mn concentrations were related to anthropogenic activity, and mainly local pollution originating from transportation as well as emissions of pollutants from distant industrial centers, conditioned by atmospheric conditions such as the wind direction. They formulated that pollution indices did not confirm the effect of distance from the road on the soil enrichment of heavy metals.

The findings of Mazurek et al. [44] are less comparable with ours in terms of concept. The background values used for calculating the indices were obtained from on-site bedrock metal contents, which operationally differed from those we applied from Håkanson [29] for worldwide

comparison and Czarnowska [32] for local regulations. On the basis of Pb geo-accumulation indices, we found that the whole park was characterized by an $I_{geo}$-Pb of 0, which is class 0 (zero, no contamination). The values of these indices were found to be site-specific, since the pattern decreased progressively according to the wind direction (Figure 1). A similar trend was also identified for $I_{geo}$-Ni, but cadmium was much less variable. We determined the total of metals within the different layers (5 to −50 cm) for each sampled site (Table 1) to eliminate seasonal year-to-year variations and hence quantifying possible metal deposition. Next, we did not use on-site bedrock metal contents as background values to avoid the alteration of $I_{geo}$ indices due to pedo-geochemical factors. This approach is consistent with the Swedish report [21] stating that, in practice, it is often difficult to distinguish clearly between the pedo-geochemical and the anthropogenic fraction of the background content of soils. The trend discrepancy observed for Pb and Ni versus Cd could be ascribed to the first two metals as they are more anthropogenic of origin. The slightly elevated Cd amounts indicate an unnatural source. The use of two different backgrounds for $I_{geo}$ calculation resulted in a consistent evaluation, showing that the KNP is not threatened by contamination.

We assessed Pb, Cd, and Ni effects and their relevant individual ecological risk (ERI) as well as potential (joint) ecological risk (PERI) indices to determine the threat level posed to the KNP. The ERI ranges observed for the Håkanson background values, i.e., 2.50–14.96, were narrower compared to the Czarnowska values, i.e., 1.28–47.52. Lead and cadmium were identified as significantly involved in shaping these indices, altering the grades (Tables 4 and 8) from moderate to very high risk. The toxicity response factors (Tr) outlined some specific metal features that seem to be more highly magnified for Cd (Tr = 30), than for Pb and Ni (Tr = 5 for both). The direct environmental toxicity of Cd is well-known, and this awareness is worth considering when evaluating natural as well as anthropogenic ecosystems.

A much more social-sensitive index, PERI, used for analyzing the overall contamination threat assessment of the whole KNP, revealed values varying within the range of 26.02 and 59.48. The ground evaluation resulted in a grading of moderate ecological risk (30 < PERI < 60). The high share of either Cd or Pb ERI values in the PERI was responsible for the moderate grade. Next, Cd could be more easily identified in this process when using the Czarnowska background value (Cd: 0.18 mg kg$^{-1}$) and the magnification related to the toxicity response factor. Notably, our findings revealed a traceable three-step geochemical referencing process should be used to evaluate heavy metal concentrations: (1) background value, (2) type of heavy metal, and (3) toxicity response factor. Specifically, using on-site bedrock metal content as the background value may limit the comparison of indices, even at the country level. Hence, our results support both local as well as worldwide referencing, mostly for natural forest ecosystems in terms of Pb, Cd, and Ni.

## 6. Conclusions

Despite the harmful external factors (proximity to the largest city in Poland, two highways, and two environmentally harmful industrial plants), Kampinoski National Park (Poland) could be considered a geochemical referencing forest ecosystem for lead, cadmium, and nickel levels.

The high share of Pb, Cd, and Ni in the −5 and −10 cm layer in the whole (5 to −50 cm) sampling depth indicates that: (1) the 5–0 cm layer should be considered for evaluation of ephemeral metal accumulation/contamination and (2) the layer −5 to −10 cm layer should be the focus when considering the perennial traceability of metal accumulation/contamination.

To enable any comparability of data between different types of soils as well as regions in the world, we recommend forest ecosystem managers to apply the sampling range adopted in this study: from litter and from depths of −5 to −10, −25 to −30, and −45 to −50 cm. These ranges allow the comparison of data and can be used to track the process of element migration in the soil layers, which is important for data interpretation.

The use of geochemical indices when assessing the contamination status of various ecosystems (natural and strongly anthropogenic) is recommended to enable a worldwide comparison, instead of using comparisons based on metal contents alone.

**Author Contributions:** P.R., J.D., Ł.T., and P.P. conceived the ideas. P.R., J.D., and M.K. designed the methodology. P.R. and M.K. conducted the field work. J.D. and A.A. performed chemical analyses. All authors prepared the manuscript. All authors have read and agreed to the published version of the manuscript.

**Funding:** The research was completed as part of a research projects co-financed in 2018 from the PGL LP Forest Fund under the titles: Environmental transformations after windbreak and assessment of microsuccession of biotopes of organisms colonizing fallen trees—stage I, and the research project Genetic characteristics and biological diversity of old tree stands with Scots pine in Kampinoski National Park—stage II. Publishing of manuscript was co-supported by Minister of Science and Higher Education in the range of the program entitled Regional Initiative of Excellence for the years 2019–2022, Project No. 005/RID/2018/2019/nauki leśne/.

**Acknowledgments:** The authors would like to thank anonymous Reviewers for their valuable comments and remarks and MDPI English Editing team for correction of English.

**Conflicts of Interest:** The authors declare no conflict of interest. The funders had no role in the design of the study; in the collection, analyses, or interpretation of data; in the writing of the manuscript, or in the decision to publish the results.

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
