# Peer review of "Geochemical Referencing of Natural Forest Contamination in Poland"

_forests, doi:10.3390/f11020157_

Round 1

Reviewer 1 Report

General comments

Many studies focus to the study of contaminated soil in different areas. The subject of the study is suitable for the journal of Forests. The article is compact and it is easy to follow. 

I have only two minor comments:

English language needs minor spell checking with a native speaker. In the Table 5 from the axis Y the studied parameter is missing. Please correct for the following: Percentage share of Pb.

Author Response

1. Authors have considered the suggestions of the Reviewer.

2. Manuscript was checked by MDPI English Editing team

3. The Table 5 was corrected.

4. We would like to thank for review

Reviewer 2 Report

The paper is acceptable for publication in Forests, since a great nuber of new data (?) -please insert sampling sites and periods on the map at Fig. 1.- is provided on the present Pb, Cd and Ni pollution (enrichment) in soil of Natural Forest Park in Poland. Geochemical indices for assessing the status of contamination (natural and anthropogenic) were also studied. However, some critism need to be taken into consoderation. Digestion in 6M HCl with heating the slurry in sand-bath at 140 oC does not give the total metal content (please rivise the title of 3.1. to: Chemical analysis of pseudo total HM content in soils. Authors should point this in the manuscript. Moreover, do authors have data on the particle size of the soils? If so, they should include this data to the text. The value of 9.5% for the average RSD for all metals is high. Perhaps, there are some problems concerning the applied analytical method of the soils. Please revise.

Author Response

1. Reviewer: Digestion in 6M HCl with heating the slurry in sand-bath at 140 oC does not give the total metal content (please rivise the title of 3.1. to: Chemical analysis of pseudo total HM content in soils. Authors should point this in the manuscript.

Authors: Authors have considered the suggestions of the Reviewer. All expressions dealing with the total content have been rewritten to pseudo-total content.

2. Reviewer: Moreover, do authors have data on the particle size of the soils? If so, they should include this data to the text.

Authors: Data has been included to the text ((Section 3.1).

3. Reviewer: The value of 9.5% for the average RSD for all metals is high. Perhaps, there are some problems concerning the applied analytical method of the soils. Please revise.

Authors: Authors are highly indebted to the Reviewer for outlying this mistake. The value was corrected to 0.95%.

4. Authors: We would like thank for review